# Multicomponent Exercise Program for Improvement of Functional Capacity and Lipidic Profile of Older Women with High Cholesterol and High Triglycerides

**DOI:** 10.3390/ijerph182010731

**Published:** 2021-10-13

**Authors:** Luis Leitão, Moacir Marocolo, Hiago L. R. de Souza, Rhai André Arriel, João Guilherme Vieira, Mauro Mazini, Teresa Figueiredo, Hugo Louro, Ana Pereira

**Affiliations:** 1Sciences and Technology Department, Superior School of Education, Polytechnic Institute of Setubal, 2910-761 Setúbal, Portugal; teresa.figueiredo@ese.ips.pt (T.F.); ana.fatima.pereira@ese.ips.pt (A.P.); 2Life Quality Research Centre, 2040-413 Rio Maior, Portugal; 3Post Graduate Program in Physical Education, Federal University of Juiz de Fora, Juiz de Fora 36036-900, Brazil; isamjf@gmail.com (M.M.); hlrsouza@gmail.com (H.L.R.d.S.); rhaiarriel@bol.com.br (R.A.A.); joaoguilhermevds@gmail.com (J.G.V.); personalmau@hotmail.com (M.M.); 4Graduate Program in Physical Education, Sudamerica Faculty, Cataguases 36774-552, Brazil; 5Sports Science School of Rio Maior, Polytechnic Institute of Santarém, 2040-413 Rio Maior, Portugal; hlouro@esdrm.ipsantarem.pt; 6Research Center in Sports Sciences, Health Sciences and Human Development (CIDESD), 5000-801 Vila Real, Portugal

**Keywords:** older women, multicomponent exercise program, detraining, dyslipidemia

## Abstract

Background: Physical inactivity is a primary cause of most chronic diseases. In addition, the negative effects of aging, physical inactivity and dyslipidemia are risk factors for cardiovascular diseases of older women. Exercise is considered fundamental for the treatment and prevention due to the benefits in the health of this population, but detraining periods after exercise can reverse them. Multicomponent exercise (ME) is a combined method of aerobic and resistance training that can improve the lipidic profile of older women with high cholesterol and triglycerides. Methods: Seventeen older women (EG: 65.3 ± 4.7 years, 1.52 ± 4.12 m) followed a supervised ME program of nine months and three months of detraining (DT), and fifteen older women (CG: 66.4 ± 5.2 years, 1.54 ± 5.58 cm) continued their daily routine, without exercise. Total cholesterol (TC), triglycerides (TG), blood glucose (GL) and functional capacity (FC) were evaluated at the beginning and at the end of the program and after three months of DT. Results: ME program improved (*p* < 0.05) lipidic profile: GL (−15.6%), TC (−15.3%), TG (−19.3%) and FC: agility (−13.3%), lower body strength (27.78%), upper body strength (26.3%), cardiorespiratory capacity (11.2%), lower body flexibility (66.67%) and upper body flexibility (85.72%). DT declined the lipidic profile and FC (*p* < 0.05). Conclusion: Lipidic profile and functional capacity can be improved with nine months of ME. Besides the negative effects of DT, three months were not enough to reverse the benefits of exercise in older women with high values of TG and TC.

## 1. Introduction

Biological aging is linked with morphological and biochemical modifications that increase the risk of developing cardiovascular diseases (CVD). Since CVD are the major cause of death among patients, its prevention should be based on the control of their risk factors such as hypertension, dyslipidemia, obesity and/or diabetes [1,2,3,4,5].

In addition, physical inactivity is related to a higher prevalence of most CVD risk factors, including high values of blood glucose, total cholesterol (TC), triglycerides (TG), high blood pressure, metabolic syndrome and obesity [3,6]. Dyslipidemia (levels that are either higher or lower than normal range for blood fats such as TG and cholesterol, and uncontrolled serum blood levels of glucose have been shown to have enormous impact on CVD risk in women compared to men, due to menopause that accentuates the negative changes in body composition and metabolic profile [4,5,6,7].

Dyslipidemia is one of the leading risk factors for diabetes and a significant cause of mortality and morbidity in the world. For example, according to the National Diabetes Observatory, in 2018, the estimated prevalence of diabetes in the Portuguese population aged between 20 and 79 years (7.7 million individuals) was 13.6%, i.e., more than 1 million people in this age group have diabetes. Moreover, more than a quarter of people aged 60–79 have diabetes and the estimated annual average direct and indirect cost of dyslipidemia is 740 million euro.

Nevertheless, during the COVID-19 pandemic, the number of hospitalizations related with acute diabetes conditions has also increased and thousands of consultations and treatments have been postponed, which in the medium and long term could have serious consequences for patients’ health. During the pandemic, diabetes was considered one of the chronic diseases that could justify missing work and it was advised that diabetics over 60 years of age should comply with a strict quarantine to avoid the risk of serious complications and death [8].

Lifestyle interventions are highly recommended for older adults, including regular physical activity; however, there has been limited research into testing lifestyle intervention effects on physical activity and dyslipidemia in older adults with diabetes over time.

For health improvement, exercise is one of the main strategies for improvement of health in quality of life in older women. Among the wide array of exercises, aerobic, resistance, high intensity interval (HIIT) and multicomponent training (ME) have been described as effective interventions to attenuate age-related dysfunctions and promote health and wellness for older women [2,4,6,9,10]. Many studies shown that exercise improves metabolic and immune functions by improving lipidic profile and inflammatory markers, as well as improves functional capacity (FC), cardiorespiratory fitness and muscular strength [2,4,11,12]. According to Albarrati et al. [13] systematic review low and moderate intensity exercise promote reduction on TC and LDL levels in dyslipidemia population, and a combined aerobic and resistance exercise training may promote additional benefits [14].

Nowadays, exercise programs in the community for individuals with specific characteristics such as age and/or diseases, such older women, are interventions suggested to increase physical activity due to their enormous applicability than individual interventions [1,2,15].

Although exercise research on older women has been valuable and an important factor in promoting exercise in the public health domain [16] and reducing inactivity and detraining periods (DP) [2,3,17], it is still unclear what are the effects of exercise and detraining periods (DP) on metabolic health and functional capacity of older women with chronic health conditions such as dyslipidemia [13]. Therefore, assessment of exercise and detraining effects in older women is an important area to understand [1,2,3], and it is more important in specific elderly population with dyslipidemia. Thus, this study aimed to analyze the effects of a nine-month multicomponent exercise program and three months of exercise cessation on lipidic profile and functional capacity of older women with levels of triglycerides and total cholesterol above normal range.

## 2. Materials and Methods

### 2.1. Sample and Ethical Procedures

In total, 32 functionally independent Caucasian women volunteered for this study. The adopted exclusion criteria were: (a) have already engaged in any physical activity program; (b) have any osteoarticular dysfunction that could interfere with the execution of the proposed exercises; (c) presence of heart problems where the exercise prescription injures the health of the older, and (d) medical contraindication. The inclusion criteria were: (a) have values above normal levels of triglycerides (>150 mg/dL) and total cholesterol (>200 mg/dL). All of volunteers underwent a medical evaluation prior starting the experimental protocol program. The study procedures were approved by the local institutional ethical committee for human experiments and were performed in accordance with the Declaration of Helsinki. In addition, all subjects signed an informed consent form before data collection. Additionally, volunteers were advised to maintain their previous lifestyle throughout the study, including dietary patterns and physical routines. Coffee, tea, alcohol and tobacco consumption and strenuous exercise were prohibited for 24 h before experimental procedures.

### 2.2. Experimental Design

Volunteers were divided in two groups: experimental (EG: *n* = 17, 65.3 ± 4.7 years, 1.52 ± 4.12 m), which engaged in a nine-month multicomponent exercise training program followed by a three month detraining period and the control group (CG: *n* = 15, 66.4 ± 5.2 years, 1.54 ± 5.58 m), which performed no training program (Table 1).

All measurements were performed by the same experienced researcher and instruments and under the same environmental conditions (10:00 h–12:00 h; 23–24 °C; 55–65% humidity) and occurred in three moments: prior of the multicomponent program (BE), at the end of the multicomponent exercise program (BD) and post three-months of detraining period (AD).

#### 2.2.1. Multicomponent Exercise Training Program (ME) 

ME consisted in 45 min, twice a week over nine consecutive months (total of 86 sessions). The program was conducted by a physical education specialist in training for older adults and followed ACSM guidelines for exercise prescription [18,19]. The training sessions consisted of aerobic and resistance training, with appropriate music for the activity, age, and interests of volunteers, structured as: 

(1) 5–8 min of global warm-up activity, including slow walk, calisthenics, and stretching exercises; 

(2) 15–25 min of cardiorespiratory workout (aerobics choreography with moderate intensity), with intensity maintained at 2–3 of the adapted Borg Rating of Perceived Exertion scale (BORG-CR10 RPE) in the first month and gradually increased up to 4–5 in the adapted Borg RPE. 

(3) 15–20 min of resistance training with exercises performed in a circuit, involving exercises for the upper (e.g.,: arm raise; bicep curl; triceps extension) and lower body (e.g.,: chair squat; air squat; calf raises; standing knee flexion), exercises for agility, mobility, and coordination (e.g.,: walking heel to toe; changing direction during walking; obstacle crossing; ball dribble; single limb stance; back leg raises), with a 20 to 30 s rest period between sets, alongside with social interaction among volunteers. Participants performed the weight resistance training using their own body weight (open and closed kinetic chain exercises) and elastic bands with three different tension levels (e.g., chest pull; lateral raise; squats; leg press) according to the adapted Borg scale (CR10—RPE) from 2–3 in the first month and gradually increased up to 4–5. Training intensity was progressive, especially in the first month of training to allow a proper familiarization with the exercises and the correct and safe technique of execution and breathing. The series and repetitions were increased month after month from 2 to 4 series and from 16 to 30 repetitions; 

(4) 5–10 min of relaxation techniques and stretching for the upper and lower body. Static and dynamic stretching techniques were included in flexibility training.

#### 2.2.2. Detraining Period (DP)

DP consisted of three consecutive months after ME, matching summer holidays. All volunteers were instructed to keep their normal lifestyles, including dietary patterns and physical routines, and to avoid any type of systematic exercise. They were frequently and systematically contacted to ensure that they were fulfilling DP requirements. The testing assessment procedures after DP were collected at the same conditions as for the ME.

#### 2.2.3. Lipidic Profile

To measure total cholesterol (TC, mg/dL), triglycerides (TG, mg/dL) and blood glucose (GL, mg/dL) were used a Cobas Accutrend Plus (Roche Diagnostics GmbH, Mannheim, Germany) in accordance with the procedures of Diabetes Atlas Committee (DAC). To carry out blood collection, we used the pen puncture Accu-Chek Softclix^®^ Pro and their lancets, graded from 1 to 3 in increasing degree of penetration depth, on the distal phalanx palmar of the third finger of the right hand.

#### 2.2.4. Functional Capacity Battery Test 

Consisted in Senior Fitness Test [20]. This protocol comprises six motor tests: upper body strength (UBS; arm curl), lower body strength (LBS; 30 s chair stand), upper body flexibility (UBF; back scratch), lower body flexibility (LBF; chair sit-and-reach), agility/dynamic balance (2TUG; 8-foot up-and-go), and aerobic capacity (6MWT; 6 min walk test).

### 2.3. Statistical Analysis 

Data analysis was made with SPSS 19.0 for Windows (SPSS Inc., Chicago, IL, USA). Descriptive procedures of central tendency and dispersion were used to characterize the variables values and the normality of our sample was verified by the Shapiro-Wilk test. Using delta percentage of the values of each variable, an independent Student *t*-test or Mann-Whitney test was performed to comparison between groups. The delta percentage (∆%) was calculated via the standard formula: ∆% = [(posttest score − pretest score)/pretest score] × 100. We used separate two-way ANOVA mixed model to compare, within and between groups, the mean values of each variable over the time, followed by the post-hoc Bonferroni test. The sphericity assumption was verified through the Mauchly’s test. The meaningfulness of the outcomes was estimated through the effect size (ES, Cohen’s d, means divided by the pooled standard deviation) classified as small (≤0.2), moderate (about 0.5) and large (≥0.8). For all statistical procedures the statistical significance accepted was *p* ≤ 0.05.

## 3. Results

The attendance rate was 87% for the subjects of EG. With ME the volunteers improved their lipidic profile and FC (*p* < 0.05), compared to CG (Table 2). Three months of detraining reduced the benefits of exercise (*p* < 0.05) but, compared to baseline values only GL benefits were reversed (Table 2 and Table 3).

Nine months of ME resulted in reductions in TC, TC, and GL, and improvements in all FC variables (*p* < 0.05).

## 4. Discussion

Our study had the objective of analyzing in long term the effect of multicomponent exercise and exercise cessation in older women with high TC and high TG. The main finding was that nine months of ME improved lipidic profile and FC near to normative values, and three months of DP reduced these benefits but were not strong enough to eliminate them compared to baseline values. These effects represent important health gains in clinical terms and in patients’ cardiovascular morbidity and mortality.

The lipidic profile of our older women improved in all variables, GL (%), TC (%) and TG (%) after ME with similar outcomes to other studies with the same or different type of exercise [7,10,11,21,22,23,24,25,26]. 

Matos et al., [16] with 12 weeks of HIIT reported improvements of 15.1% in TC, 24.3% in TG and 7% in GL and refer that two session a week are feasible and effective to induce relevant improvements in older women. Kang et al., [27] with 12 weeks of aerobic exercise obtained 8.8% in TC, 34.4% in TG and 5% in GL. In a seven-year study with aerobic exercise Zajac-Gawlak, et al. [28] observed 16.6% and 9.4% improvement in TG and GL, respectively, referring that increases in daily steps/day is related to a lower risk of metabolic syndrome. Using RT, Cunha, Ribeiro, Nunes, Tomeleri, Nascimento, Moraes, Sugihara, Barbosa, Venturini and Cyrino [4] compared low volume and high volume and stated that booth improved lipidic variables but high volume shown better results, with improvements of 14.7% in TC, 16.6% in TG and 11.1% in GL. Arnarson et al., [29] with 12 weeks of RT resulted in 12.8% improvement in TC and 12.1% in TG. With nine month of ME Marques et al. [15] observed improvement in TG (5.1%) and Mendes et al. [1] reported benefits of 13.1% in TC, 25.1% in TG and 14.9% in GL. 

Our lipidic improvement can be explained from the combined type of exercise of ME. Regular aerobic exercise promotes improvements in TC, TG and GL [21,25,27,30] and enhances oxidative enzymes that are associated to better insulin sensitivity [30]. RT training is also powerful for changing the level of lipoproteins [20], to improve the decline in myosin chain and increase protein synthesis rate leading to enhanced glucose oxidation and a better insulin sensitivity, and to improve metabolic biomarkers in elderly women but are depended on volume, intensity and initial level of the subjects, since participants with higher GL and TG values can obtain better results [21,22,30]. Additionally, RT can increase the ability of skeletal muscle to use fat, reducing the plasma lipids levels [7]. TG improvements also can be explained through reduction of BF% not weight, since losing weight in this population may not be beneficial to their health [31,32], and from energy expenditure of exercise that is the primary determinant of postprandial TG reduction. The mechanism that can explained this post prandial TG reduction is the increased muscle lipoprotein and from a better clearance from the circulation after exercise since the blood flow increased with exercise [33]. According to Nilsson, et al. [34] and Wewege et al. [21] increased amounts of physical activity should be recommend to promote a favorable lipidic health profile [11,30] and should have more than 12 weeks.

The improvements observed in FC were similar to other studies [2,35,36,37]. Boukabous et al. [9] reported that VO2 peak improved after 2–4 weeks of exercise and cardiorespiratory fitness improved after 8 weeks [10]. Leitão et al., [38] observed a 10.87% improvement in aerobic capacity after 9 months of ME with older women with lipidic disorder. With 12 weeks of aerobic exercise Prusik et al. [10] reported a 7.7% improvement in upper body strength, 4.8% in lower body strength, 10.2% in agility and 28.5% in lower body flexibility. The cessation of exercise resulted in declines in FC and lipidic profile, although were not enough to reverse to baseline values [2,6,38,39,40]. Padilha et al. [41] with 12 weeks of detaining after 12 weeks of RT observed declines in FC but not significantly compared to baseline value. In other studies, with older women without dyslipidemia DT periods were enough to reverse exercise benefits, such Leitão et al., [38] with the same DP period after 9 months of ME reported −8% and −11% in TC and TG, respectively, indicating that was enough to reverse all exercise benefits in older women without dyslipidemia. Prusik et al. [10] with 6 months of detraining after 12 weeks of aerobic exercise reported that TC and TG returned to baseline values. These modifications had a negative impact in the health of older adults, especially in older women with dyslipidemia, and contribute to increase sedentarism and CVD risk factors [1,2].

The present study has some limitations, such as not controlling the subjects’ dietary regimens during training and detraining periods. However, considering the continuity of activities, since all volunteers followed their routines, without modifications, this point may not have been of great influence throughout the experimental protocol.

## 5. Conclusions

Multicomponent exercise promotes significant improvements in FC and in the health of older women with high TC and TG since booth were near to reaching normative values. Three months of exercise cessation were not enough to reverse lipidic profile to baseline values in older women.

## Figures and Tables

**Table 1 ijerph-18-10731-t001:** Subject’s anthropometric characteristics.

Variable	Group	BaseLine	Detraining Period (3 Months)	
		Before Exercise (BE)	Beginning of Detraining (BD)	After Detraining (AD)
Body weight (kg)	EG	73.38 ± 10.28	71.98 ± 10.25	72.39 ± 10.27
	CG	71.11 ± 9.94	71.23 ± 10.03	71.18 ± 9.89
Height (cm)	EG	153.00 ± 4.12	153 ± 4.12	153 ± 4.12
	CG	155.47 ± 5.58	155.47 ± 5.58	155.47 ± 5.58
BF (%)	EG	31.35 ± 4.24	30.75 ± 4.24	30.92 ± 4.24
	CG	29.59 ± 5.11	29.64 ± 5.14	29.62 ± 5.12

Data presented are mean ± SD, before detraining (BD) and after detraining (AD) of body fat percentage (BF%), body mass index (BMI) and body weight (kg).

**Table 2 ijerph-18-10731-t002:** Delta differences of parameters after multicomponent exercise program and detraining period.

	∆ BE and BD			∆ BE and AD			∆ BD and AD		
Variable	CG	EG	*p*-Value	CG	EG	*p*-Value	CG	EG	*p*-Value
Body weight (%)	0.16 ± 1.01	−1.94 ± 0.96	<0.01	0.12 ± 0.94	−1.4 ± 0.84	<0.01	−0.04 ± 0.73	0.58 ± 0.42	<0.01
Height (%)	0.00 ± 0.00	0.00 ± 0.00	1.00	0.00 ± 0.00	0.00 ± 0.00	1.00	0.00 ± 0.00	0.00 ± 0.00	1.00
BMI (%)	0.16 ± 1.01	−1.94 ± 0.96	<0.01	0.12 ± 0.94	−1.37 ± 0.84	<0.01	−0.04 ± 0.73	0.58 ± 0.42	<0.01
BF (%)	−0.21 ± 0.94	−2.48 ± 0.75	<0.01	0.46 ± 1.39	−1.87 ± 0.77	<0.01	0.68 ± 1.61	0.62 ± 0.82	0.664
TG (%)	−1.20 ± 4.70	−16.17 ± 5.61	<0.01	−0.27 ± 4.97	−11.45 ± 5.85	<0.01	1.00 ± 3.78	−5.79 ± 6.02	0.013
TC (%)	−0.17 ± 1.51	−18.72 ± 3.51	<0.01	−1.18 ± 3.82	−10.35 ± 2.42	<0.01	−1.00 ± 3.76	−10.47 ± 4.00	<0.01
GL (%)	0.87 ± 3.48	−15.92 ± 8.10	<0.01	1.31 ± 3.71	−1.39 ± 4.72	0.104	0.51 ± 4.05	−18.11 ± 10.80	<0.01
LBS (%)	2.41 ± 5.29	30.54 ± 13.67	<0.01	0.92 ± 8.56	14.68 ± 12.36	<0.01	−1.12 ± 10.64	−12.01 ± 4.77	<0.01
UBS (%)	1.87 ± 7.53	38.39 ± 16.31	<0.01	0.65 ± 10.56	23.80 ±18.37	<0.01	−1.10 ± 8.40	−10.16 ± 10.87	0.014
2TUG (%)	0.72 ± 1.76	−10.11 ± 7.49	<0.01	0.32 ± 2.54	−6.84 ± 7.05	<0.01	−0.41 ± 0.82	3.83 ± 5.33	<0.01
6MWT (%)	−0.47 ± 4.02	12.73 ± 9.68	<0.01	−1.08 ± 2.79	5.89 ± 5.59	<0.01	−0.54 ± 2.81	−5.74 ± 5.05	<0.01
UBF (cm)	0.27 ± 1.49	5.41 ± 2.40	<0.01	0.00 ± 1.73	4.65 ± 2.47	<0.01	−0.27± 1.28	−0.77 ± 0.44	0.234
LBF (cm)	0.47 ± 1.06	7.59 ± 2.96	<0.01	0.07 ± 1.10	6.88 ± 3.26	<0.01	−0.40 ± 0.99	−0.71 ± 1.80	0.656

Data presented are mean ± SD; BE: Before Exercise; BD: Beginning of Detraining; AD: After Detraining; EG: exercise group (*n* = 17); CG: control group (*n* = 15); BMI: Body mass index; TG: Triglycerides; TC: Total cholesterol; GL: Blood glucose; LBS: Lower body strength; UBS: Upper body strength; 2TUG: Agility/dynamic balance; LBF: Lower body flexibility; UBF: Upper body flexibility; 6MWT: aerobic endurance six-minute walk test.

**Table 3 ijerph-18-10731-t003:** Effects of the multicomponent exercise program and detraining in lipidic health profile and functional capacity of older women with dyslipidemia.

Variable	CG							EG						
	BE	BD	AD	BE vs. AD				BE	BD	AD	BE vs. AD			
				95% Confidence Interval		ES	*p*				95% Confidence Interval		ES	*p*
				Lower	Upper						Lower	Upper		
TG (mg/dL)	185.07 ± 35.73	181.87 ± 28.93	183.33 ± 27.96	−6.74	10.21	−0.06	1.00	196.41 ± 29.73	164.06 ± 23.28 *	172.76 ± 20.43 ^+^	15.69	31.61	−0.96	0.01
TC (mg/dL)	211.33 ± 21.15	211.00 ± 21.71	208.60 ± 20.21	−1.63	7.10	−0.14	0.36	221.35 ± 21.85	180.24 ± 22.27 *	198.71 ± 22.59 ^+^	18.54	26.75	−1.05	0.01
GL (mg/dL)	86.33 ± 4.32	87.07 ± 4.95	87.40 ± 4.14	−3.60	1.47	0.26	0.88	92.47 ± 6.12	77.76 ± 9.33 *	91.35 ± 9.53 ^+^	−1.27	3.50	−0.14	0.73
LBS (repetitions)	15.93 ± 2.12	16.33 ± 2.41	16.13 ± 2.83	−1.21	0.81	0.08	1.00	17.35 ± 3.66	22.29 ± 3.51 *	19.65 ± 3.35 ^+^	−3.25	−1.34	0.67	0.01
UBS (repetitions)	17.87 ± 3.50	18.07 ± 3.17	17.73 ± 2.76	−1.01	1.28	−0.04	1.00	17.18 ± 3.75	23.29 ± 3.72 *	20.76 ± 3.23 ^+^	−4.66	−2.51	1.06	0.01
2TUG (s)	5.64 ± 0.61	5.68 ± 0.62	5.66 ± 0.61	−0.23	0.20	0.03	1.00	5.75 ± 0.52	5.16 ± 0.60 *	5.35 ± 0.55 ^+^	0.20	0.61	−0.77	0.01
UBF (cm)	−7.13 ± 3.44	−6.87 ± 3.85	−7.13 ± 3.60	−1.41	1.41	0.00	1.00	−7.06 ± 3.98	−1.65 ± 3.97 *	−2.41 ± 4.03 ^+^	−5.97	−3.32	1.20	0.01
LBF (cm)	−2.80 ± 4.62	−2.33 ± 4.42	−2.73 ± 4.50	−1.70	1.57	0.02	1.00	−2.65 ± 4.27	4.94 ± 3.07 *	4.24 ± 3.25 ^+^	−8.42	−5.35	1.87	0.01
6MWT (m)	575.00 ± 42.30	572.33 ± 47.20	568.67 ± 42.40	−8.20	20.87	−0.15	0.83	562.06 ± 69.55	629.12 ± 56.49 *	592.94 ± 60.49 ^+^	−44.53	−17.23	0.49	0.01

Data presented are mean ± SD; before multicomponent training (BE), before detraining period (BD) and after detraining (AD) of Triglycerides (TG), Total cholesterol (TC), Blood glucose (GL), Lower body strength (LBS), Upper body strength (UBS), Agility/dynamic balance (2TUG), Lower body flexibility (LBF), Upper body flexibility (UBF), aerobic endurance six-minute walk test (6MWT); EG: exercise group (*n* = 17); CG: control group (*n* = 15); Effect Size (ES); * *p* < 0.05, significant improvements after training period of multicomponent training program; ^+^ *p* < 0.05, significant decreases after detraining period multicomponent training program.

## Data Availability

The data presented in this study are available on request from the corresponding author.

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
