# Peer review of "Multicomponent Exercise Program for Improvement of Functional Capacity and Lipidic Profile of Older Women with High Cholesterol and High Triglycerides"

_ijerph, 2021, doi:10.3390/ijerph182010731_

Round 1

Reviewer 1 Report

This is an interesting study that shows maintenance of the effect of physical activity during 3 months of inactivity. Thank you for the opportunity to read this article. Here are few comments.

How did you decide to choose 3 months of inactivity? Did you have any premises? This is interesting because the COVID-19 lockdown was longer. Do you have any conclusions on how to predict a safe inactivity period after a certain period of regular exercising?

Can add some insight on the patients' view on detraining? What was their willingness to continue the training? I also find it unethical to stop training if they were willing to continue. They were discouraged to use a beneficial intervention. Can you comment on this?

How did you calculate BMI? Obese is defined with BMI over 30, but in Table 1, values are over 150.

Can you compare values from Table 1? I would like to know if there are any significant differences between the groups. Also, you can add height to Table 1 as it is mentioned in Table 2.

What was the energy expenditure per exercise session? Can you compare the energy expenditure per week with other publications that examined the impact of detraining?

Language review is necessary. Here below there are examples of language issues in the paper:

Capitalisation in the abstract in lines 26-27 is overused. Cholesterol, triglycerides, blood glucose should be spelt with small letters.

Shorts like weren´t should not be used in the scientific literature.

The sentence in lines 40-42 is not clear. What does “Since CVD last one are 40 the major cause” mean? The rest of the sentence is singular which further blurs clarity.

Sort out abbreviations. Use them when there are 3 or more. Once the term is abbreviated, it should be used as an abbreviation in the rest of the manuscript. See the example BP (line 45) – it is not necessary; triglycerides in line 46 should be abbreviated, cardiovascular disease in line 48 should be abbreviated, etc. In addition, the abbreviations in this paper are overused making the text hard to read.

Author Response

Dear reviewer,

We thank all the reviewers for their interest and constructive remarks regarding our manuscript: “Multicomponent exercise program for improvement of functional capacity and lipidic profile of older women with dyslipidemia”.

We have revised the manuscript according to their recommendation. All answers and modifications are following in this reply and in the manuscript.

Luis Filipe Leitão

How did you decide to choose 3 months of inactivity? Did you have any premises? This is interesting because the COVID-19 lockdown was longer. Do you have any conclusions on how to predict a safe inactivity period after a certain period of regular exercising?

            R.: Three months were chosen because it seems that, after this period, the decline effects of detraining are less accentuated. According to the literature after eight weeks of detraining the negative effects start to be significant (Padilha et al., 2015; doi: 10.1007/s11357-015-9841-6). In our opinion COVID-19 lockdown may cause more negative effects than our detraining period because the population stayed in their homes almost all the time and the inactivity time increased.

Can add some insight on the patients' view on detraining? What was their willingness to continue the training? I also find it unethical to stop training if they were willing to continue. They were discouraged to use a beneficial intervention. Can you comment on this?

            R.: The exercise program was a community program discontinued during summer time and vacations. Most of the participants stated that they feel the difference (detrimental effects) doing their daily living activities after two months of interruption (reported when the program re-started). This is one more evidence that exercise should not be stopped and be continuous through life time, aiming at keeping the physical fitness of older 

How did you calculate BMI? Obese is defined with BMI over 30, but in Table 1, values are over 150. Can you compare values from Table 1? I would like to know if there are any significant differences between the groups. Also, you can add height to Table 1 as it is mentioned in Table 2.

            R.: Sorry for this mistake. We have modified accordingly and changed BMI to Height in the table 1.  There were no differences between groups in baseline values.

What was the energy expenditure per exercise session? Can you compare the energy expenditure per week with other publications that examined the impact of detraining?

            R.: We have not measured energy expenditure during exercise program. Our sessions were controlled by rate of perceived exertion, described in the text.

Language review is necessary. Here below there are examples of language issues in the paper:

Capitalisation in the abstract in lines 26-27 is overused. Cholesterol, triglycerides, blood glucose should be spelt with small letters. Shorts like weren´t should not be used in the scientific literature. The sentence in lines 40-42 is not clear.

R.: The language was edited by an English native speaker.

What does “Since CVD last one are 40 the major cause” mean? The rest of the sentence is singular which further blurs clarity.

            R.: Sorry for this mistake. The sentence was completely modified to: “Since CVD are the major cause of death among patients, its prevention should be based on the control of their risk factors such as hypertension, dyslipidemia, obesity and/or diabetes”.        

Sort out abbreviations. Use them when there are 3 or more. Once the term is abbreviated, it should be used as an abbreviation in the rest of the manuscript. See the example BP (line 45) – it is not necessary; triglycerides in line 46 should be abbreviated, cardiovascular disease in line 48 should be abbreviated, etc. In addition, the abbreviations in this paper are overused making the text hard to read.

            R.: We agree with this comment. All abbreviations were used or excluded accordingly.

Reviewer 2 Report

I reviewed the article: "Multicomponent exercise for improvement of functional capacity and health of older women with dyslipidemia", whose objective was not clearly stated, but according to the title, to study the effect of exercise on functional capacity and health(?), in women with dyslipidemia.

There are many published studies with good designs and extensive analytical procedures, (EXERCISE) AND ((lipoprotein) OR ((hyperlipidemia) OR (dyslipidemia)). So the contribution of the present manuscript is not significant for the knowledge of the effect on functional capacity, neither on "health" (which they did not estimate), neither on total cholesterol and triglycerides. In fact this group of authors have already published something very similar with identical designs and methodologies, also published in this same journal in 2019 (doi:10.3390/ijerph16203881), (it varied that they measured VO2, and that they were not women with cholesterol or triglycerides above the recommended levels). This, without doubting the integrity of the authors, raises uncertainties for me about the ethics of research and the ethics of scientific publication.

The manuscript presents and reflects important conceptual and methodological weaknesses.

What do the authors of the study understand by dyslipidemia or what concept do they use, is there only one type of dyslipidemia, is there only one degree of dyslipidemia, if patients have dyslipidemia, what are the different lipoprotein values, what is the atherogenic index, etc.? If patients have dyslipidemia, what are the values of the different lipoproteins, what is the atherogenic index, etc...? If they have "dyslipidemia", will the patients have a medical treatment, what treatment did they have? How long did they have evolution? It is questionable whether clinically significant conclusions can be drawn only by analyzing total cholesterol and triglycerides. Moreover, it is not correct to interpret diseases in a dichotomous way: they either have the disease or they do not have the disease. To which patients could we apply the theoretical results of this study?

Another very deficient aspect, in terms of disease or risk factors, is to make conclusions based solely on the "p" value. How many patients improved and how many did not? What is the minimum change in the concentration of total cholesterol and/or total triglycerides that is considered clinically significant? What is the variability, the minimum detectable change of the methods used? What is the sensitivity to detect change and the accuracy of the tests of the "senior fitness test battery"?

The exclusion criteria are detailed, but the inclusion criteria are not included, nor are the other comorbidities that I imagine a large proportion of the sample would present, nor the medications taken, nor the years of treatment or knowledge of the disease. How many had only "elevated" total cholesterol and how many had only triglycerides?

Other confounding factors were not controlled, such as diet or physical activity outside the program (it is not enough to say that they continued with the same habits, which will be different in winter than in summer; these are fundamental factors that without their control, the conclusions regarding lipid profile will have little validity). Where was the sample obtained from? What was its occupation? How was the sample size determined, which clearly seems insufficient? What was the criterion or the way of assigning the patients to one group or another? It is not enough to state, as the same authors do in other similar publications, that the study was approved by a local committee, what committee?  What is the registration number of this authorization?. Where and on what dates was the study carried out? (It is curious that even the environmental conditions of humidity and temperature were identical (copy-paste), from the above-mentioned study by the same authors.

Is it plausible to assume that 35 minutes of activity at a light intensity (plus 5 minutes of activation and 5 minutes of relaxation), twice a week can produce changes in the lipid profile in people who are not extremely sedentary? This is what should be developed in the introduction, which is generic and does not focus on justifying the relevance or plausibility of the object of the study, which is not aging in general (neither is the mention of COVID without any bibliographic reference), or discuss it in depth in the discussion. I miss in the introduction and in the discussion, some recognized articles of systematic reviews of the effect of physical exercise on lipoproteins and on dyslipidemias.

They should justify why they call moderate intensity exercise a subjective perception of effort 2-3 or 4-5 (RPE).

Of course, the program does not follow the ACSM guidelines (ref 15 and 16), even though the authors claim to follow them.

How in a collective exercise, is the individual load individualized in choreographed exercises? What is the intensity of the exercise (not training) of strength with self-loading and/or with elastics - what resistance of the elastic - what exercises did they perform? The study can hardly be replicated by other researchers with the data provided.

The title does not clearly reflect the content of the article, what is "health" and how did they estimate it?

There is an error in Table 1 when describing BMI.

I believe that it is not correct, with the format of the journal, to abuse to cite all the authors of a study in the discussion, if necessary, in the worst case to place the first author and abbreviated collaborators.

Author Response

Dear reviewer,

We thank all the reviewers for their interest and constructive remarks regarding our manuscript: “Multicomponent exercise program for improvement of functional capacity and lipidic profile of older women with dyslipidemia”.

We have revised the manuscript according to their recommendation. All answers and modifications are following in this reply and in the manuscript.

Luis Filipe Leitão

There are many published studies with good designs and extensive analytical procedures, (EXERCISE) AND ((lipoprotein) OR ((hyperlipidemia) OR (dyslipidemia)). So the contribution of the present manuscript is not significant for the knowledge of the effect on functional capacity, neither on "health" (which they did not estimate), neither on total cholesterol and triglycerides. In fact this group of authors have already published something very similar with identical designs and methodologies, also published in this same journal in 2019 (doi:10.3390/ijerph16203881), (it varied that they measured VO2, and that they were not women with cholesterol or triglycerides above the recommended levels). This, without doubting the integrity of the authors, raises uncertainties for me about the ethics of research and the ethics of scientific publication.

R.: We agree with this comment about studies on this topic. However, there is no study which analyzed the impact of multicomponent exercise program in functional capacity, total cholesterol and triglycerides of older women with dyslipidemia. The study design is the same of the referred article but here the participants have specific lipidic disorders (high triglycerides and high cholesterol) that may or may not affect the effects of exercise and detraining on health profiles and functional capacity in comparison with older women without this specificity, although we think age maybe the leading factor of a possible change in exercise and detraining effects.

The manuscript presents and reflects important conceptual and methodological weaknesses.

What do the authors of the study understand by dyslipidemia or what concept do they use, is there only one type of dyslipidemia, is there only one degree of dyslipidemia, if patients have dyslipidemia, what are the different lipoprotein values, what is the atherogenic index, etc.? If patients have dyslipidemia, what are the values of the different lipoproteins, what is the atherogenic index, etc...? If they have "dyslipidemia", will the patients have a medical treatment, what treatment did they have? How long did they have evolution? It is questionable whether clinically significant conclusions can be drawn only by analyzing total cholesterol and triglycerides. Moreover, it is not correct to interpret diseases in a dichotomous way: they either have the disease or they do not have the disease. To which patients could we apply the theoretical results of this study?

R.: We agree with above statement. The section regarding inclusion/exclusion criteria were rewritten in more details as well as a limitation paragraph at the end of discussion. Also, we added more details about medical record of patients.

Another very deficient aspect, in terms of disease or risk factors, is to make conclusions based solely on the "p" value. How many patients improved and how many did not? What is the minimum change in the concentration of total cholesterol and/or total triglycerides that is considered clinically significant? What is the variability, the minimum detectable change of the methods used? What is the sensitivity to detect change and the accuracy of the tests of the "senior fitness test battery"?

R.: We have added the pre- and post-values of lipid profile parameters.

The exclusion criteria are detailed, but the inclusion criteria are not included, nor are the other comorbidities that I imagine a large proportion of the sample would present, nor the medications taken, nor the years of treatment or knowledge of the disease. How many had only "elevated" total cholesterol and how many had only triglycerides?

R.: Sorry for this mistake. As we stated above, the inclusion/exclusion criteria were more detailed.

Other confounding factors were not controlled, such as diet or physical activity outside the program (it is not enough to say that they continued with the same habits, which will be different in winter than in summer; these are fundamental factors that without their control, the conclusions regarding lipid profile will have little validity). Where was the sample obtained from? What was its occupation? How was the sample size determined, which clearly seems insufficient? What was the criterion or the way of assigning the patients to one group or another? It is not enough to state, as the same authors do in other similar publications, that the study was approved by a local committee, what committee?  What is the registration number of this authorization? Where and on what dates was the study carried out? (It is curious that even the environmental conditions of humidity and temperature were identical (copy-paste), from the above-mentioned study by the same authors.

R.: We have added a limitation part in the discussion. “The present study has some limitations, such as not controlling the subjects’ dietary regimens during training and detraining periods. However, considering the continuity of activities, since all volunteers followed their routines, without modifications, this point may not have been of great influence throughout the experimental protocol.”. The multicomponent exercise program is a community program which runs from October to July every year, that has the participation of more than 120 older women that are already retired and had attended a clinical session to be allowed their participation. The criteria to assigning to the group of exercise was all the older women must have high values of triglycerides and total cholesterol and wanted to complete the exercise program, and to the control group the older women that only wanted to do the assessments sessions of the lipidic profile and functional capacity battery tests. The environmental conditions of the assessment’s sessions were the same because were made in the same time of the year and location. Although the sample size of this article is not large a power of 0.99 was reached. To perform a power statistical analysis, we used the G*Power software. Thus, a post hoc power analysis was performed using:

- Sample size of this study (32 participants);

- An effect size (ES) of 0.91 (This value was reached after we performed an average of all ES found in the exercise group inserted in table 3);

- α = 0.05;

- Statistical Test = Means: difference between two dependent means (matched pairs). 

A power of 0.99 was reached. This power set is higher than conventional value of 0.80 [DOI: 10.3758/bf03193146]. In post hoc analyses, 1 - β is computed as a function of α, the population effect size parameter, and the sample size(s) used in a study. It thus becomes possible to assess whether or not a published statistical test in fact had a fair chance of rejecting an incorrect H0 [DOI: 10.3758/bf03193146]. Therefore, we can consider that our simple size is adequate for analyses.

Is it plausible to assume that 35 minutes of activity at a light intensity (plus 5 minutes of activation and 5 minutes of relaxation), twice a week can produce changes in the lipid profile in people who are not extremely sedentary? This is what should be developed in the introduction, which is generic and does not focus on justifying the relevance or plausibility of the object of the study, which is not aging in general (neither is the mention of COVID without any bibliographic reference), or discuss it in depth in the discussion. I miss in the introduction and in the discussion, some recognized articles of systematic reviews of the effect of physical exercise on lipoproteins and on dyslipidemias.

R.: We add reference in COVID paragraph and rewrite the introduction and add two systematic reviews (Line: 85 -100)

They should justify why they call moderate intensity exercise a subjective perception of effort 2-3 or 4-5 (RPE).

R.: We clarified this part of the text. We used the modified Borg scale (CR10). Borg CR10 scale has 12 categories with values ranging from 0 to 10. Values 3 and 4 are described as "moderate" and "somewhat difficult", respectively. (Borg GA. Psychophysical bases of perceived exertion. Med Sci Sports Exerc. 1982;14:377–381)

Of course, the program does not follow the ACSM guidelines (ref 15 and 16), even though the authors claim to follow them. How in a collective exercise, is the individual load individualized in choreographed exercises? What is the intensity of the exercise (not training) of strength with self-loading and/or with elastics - what resistance of the elastic what exercises did they perform? The study can hardly be replicated by other researchers with the data provided.

R.: We rewrite the methods and add samples of exercises. To control the exercise load we used BORG-10 RPE. In choreographed aerobic exercise we used the same bpm for all and in the resistance exercise with elastic bands the older women choose the band that had the right tension level for them.

The title does not clearly reflect the content of the article, what is "health" and how did they estimate it?

R.: We have changed the title. “Multicomponent exercise program for improvement of functional capacity and lipidic profile of older women with dyslipidemia”

There is an error in Table 1 when describing BMI.

R.: Sorry for this mistake. We have modified accordingly and changed BMI to Height in the table 1.

I believe that it is not correct, with the format of the journal, to abuse to cite all the authors of a study in the discussion, if necessary, in the worst case to place the first author and abbreviated collaborators.

R.: We apologize for that. We check the citation style throughout the text.

Reviewer 3 Report

I would like to congratulate the authors for the study carried out.

My minor comments to the authors are:

  • In the title, authors must include the word “program”
  • The sample size is small. They should describe the study as a pilot study.
  • I missed inclusion criteria. Why did they only include women?
  • The authors should explain better the exercises they perform: stretching, calisthenics.
  • The authors should explain better the exercises they perform: exercises for agility, mobility, and coordination, …. Participants performed the weight resistance training using their own body weight (open and closed kinetic chainexercises) and elastic
  • Authors should include a limitation section.
  • The conclusion should delete the last sentence.
  • References: The style of the references is correct.

Author Response

Dear reviewer,

We thank all the reviewers for their interest and constructive remarks regarding our manuscript: “Multicomponent exercise program for improvement of functional capacity and lipidic profile of older women with dyslipidemia”.

We have revised the manuscript according to their recommendation. All answers and modifications are following in this reply and in the manuscript.

Luis Filipe Leitão

In the title, authors must include the word “program”

R.: We agree and the title was modified accordingly.

The sample size is small. They should describe the study as a pilot study.

R.: We thank you for your suggestion. However, we decided to perform a power statistical analysis through the G*Power software. Thus, a post hoc power analysis was performed using:

- Sample size of this study (32 participants);

- An effect size (ES) of 0.91 (This value was reached after we performed an average of all ES found in the exercise group inserted in table 3);

- α = 0.05;

- Statistical Test = Means: difference between two dependent means (matched pairs).

A power of 0.99 was reached. This power set is higher than conventional value of 0.80 [DOI: 10.3758/bf03193146]. In post hoc analyses, 1 - β is computed as a function of α, the population effect size parameter, and the sample size(s) used in a study. It thus becomes possible to assess whether or not a published statistical test in fact had a fair chance of rejecting an incorrect H0 [DOI: 10.3758/bf03193146]. Therefore, we can consider that our simple size is adequate for analyses.

I missed inclusion criteria. Why did they only include women?

R.: We apologize for that. The section regarding inclusion/exclusion criteria were rewritten in more details. Specifically, “The inclusion criteria were: a) have high values of triglycerides and total cholesterol.”. The multicomponent exercise program was only for older women.

The authors should explain better the exercises they perform: stretching, calisthenics.

The authors should explain better the exercises they perform: exercises for agility, mobility, and coordination, …. Participants performed the weight resistance training using their own body weight (open and closed kinetic chain exercises) and elastic

R.: We agree and we rewrite. Lines 290-298.

Authors should include a limitation section.

R.: We agree with this comment. We have added a limitation part in the discussion. “The present study had some limitations, such as not controlling the subjects’ dietary regimens during training and detraining periods.

The conclusion should delete the last sentence.

R.: We have deleted this sentence.

Round 2

Reviewer 2 Report

I have read the second version of the manuscript and the authors' responses to the points I made in the first review. I note that except for the correction of errors, they have not made any really important changes to the manuscript. I am not satisfied with the answers they have given either; I am trying to explain the critical, uncorrected errors (I do not know if they could be remedied with the data they have from the participants in their intervention. These critical errors, conceptual, methodological and design errors, produce a loss of interest in the work and can also produce errors of interpretation (or assumption of truths) by readers without sufficient knowledge of the "pathology" of the participants in the study.

It is always imprudent to formulate absolute truths, you say that there are no studies to see the effect of multicomponent training on lipids in older women; of course there are, for example by another group of Portuguese researchers (https://doi.org/10.1016/j.archger.2009.05.020).

The authors still do not correctly define the concept of hyperlipidemia (for them it is the increase in cholesterol "and" triglycerides above "normative values"). To support their definition, they use literature that is not specific to hyperlipidemia, but rather the same erroneous concept used by other authors who also intervened through exercise, not articles from recognized specialists in hyperlipidemia, nor from world-renowned societies (National Cholesterol Education Panel (NCEP), Adult Treatment Panel III (ATP III), European Atherosclerosis Society (EAS)...). They should keep in mind that a person who is heavier than the recommended weight is not necessarily obese; a person who has blood glucose values higher than desirable is not necessarily a diabetic. A person with less bone mineral density than desirable (normal) does not necessarily have osteoporosis. A person with less strength or muscle mass than expected does not necessarily have sarcopenia. A person who has cholesterol "or" triglycerides above the desirable values does not have dyslipidemia. There are some diagnostic criteria, some diagnostic cut-off points: there is pre-sarcopenia, there is osteopenia, there is overweight, etc. .... and they do not indicate which are the normative points they use for cholesterol and triglycerides. However, we should point out that for cholesterol, less than 200 is desirable, between 200 and 240 is borderline, and above 240 is hypercholesterolemia (ATPIII; NCEP, EAS...)..The study participants have slightly borderline values (211.33 and 221.35 mg/dL), not hypercholesterolemia (average values). Except that to maintain these borderline values, they need to take cholesterol-lowering agents, which you do not indicate whether or not you take, nor what drug you take (if they take).

There is an important bias, not stated in the document but in its response to this reviewer, since the training group were women who did want (did they have the conditions?) to participate in the training sessions, while the control group were women who did not want to participate in the training sessions. What were the reasons for not "wanting/powering" to participate?

The sample is poorly characterized, the origin of these people is not indicated, the data of the Bioethics Committee is still not provided, nor is the date or at least the code assigned by this committee to this research. In studies with clinical conclusions, neither the group mean nor the comparison of group means is sufficient: how many in each group improved and how many did not? The cut-off points for cholesterol as a risk factor are usually established in mmol/L. How many changed this value above the value that modifies the calculable risk factor?

I don't understand table 3, I want to understand that what they are trying to expose with the statistical treatment is the ANOVA, the three moments of evaluation, but independently for each of the two groups (which is not the best), but the post hoc result is not pointed out (who is different from whom? BE vs BD, BE vs AD, BD vs AD), and they do not indicate what is "ES", I want to understand that it is the effect size, which in the methodology they say it is Cohen's, but what they have done (I think) is an ANOVA, so maybe the effect size is not and should not be Cohen's but others like partial eta squared. 

They should show the values of all the physical tests that were performed, not just the delta, and compare whether there was a difference between the groups in each physical test and BMI.

The lack of clarity in the definition of hyperlipidemia, throughout the text they make hyperlipidemia synonymous with cholesterol above the desirable values (below clearly pathological values), so that they argue with articles that have studied the effect of physical exercise on hyperlipidemias (C-HDL, C-LCD, C-VDL,...) with their results on the supposed effect of exercise on cholesterol and triglycerides. For example the important reference they have introduced (ref. 13, Albarrati et al.) does not conclude as they say that physical exercise reduces cholesterol, the study concludes something different: "Tis study found that low- to moderate-aerobic exercise intensities did not reduce the levels of LDL except in few studies that have been limited to specifc populations...."

The studies included in the meta-analyses of the effect of physical exercise on hyperlipidemia, in the vast majority have studied lipoproteins; and the effects of exercise are focused on these lipoproteins, which are clearly related to cardiovascular risk, much more than cholesterol (and less so borderline cholesterol), (e.g., the effects of exercise on the lipoproteins that are clearly related to cardiovascular risk, much more than cholesterol (and less so borderline cholesterol):, (e.g.:  DOIs: 10.5551/jat.42937 , 10.1089/acm.2017.0104,   10.1038/sj.ejcn.1600784, 10.1161/JAHA.115.002014)

The description of the intervention is still insufficient; with the data provided, other researchers would not be able to replicate this study: can individual intensity be prescribed in collective choreographies? What level of resistance did the elastics have? what resistance is "adequate" for each person? was the same elastic for the leg press exercise as for the lateral raise, or for the squats...? Can the whole session described be done in 45 minutes?

In short, it is a very interesting subject and of great utility for people whose health is threatened, and whose conclusions can be applicable to these people, so the designs of these studies must be impeccable, and the conclusions we reach must be well-founded and based on clear diagnostic concepts. This is why we must be extremely responsible and thorough in our research and in the review of our peers' research.

Author Response

Dear reviewer,

We thank your interest and constructive remarks regarding our manuscript. 

We have revised the manuscript according to your recommendation. All answers and modifications are following in this reply and in the manuscript.

Thank you

Best regards

It is always imprudent to formulate absolute truths, you say that there are no studies to see the effect of multicomponent training on lipids in older women; of course there are, for example by another group of Portuguese researchers (https://doi.org/10.1016/j.archger.2009.05.020).

R: We agree, but in this article the older women have normal values of triglycerides and they do not analyze the detraining effects.

The authors still do not correctly define the concept of hyperlipidemia (for them it is the increase in cholesterol "and" triglycerides above "normative values"). To support their definition, they use literature that is not specific to hyperlipidemia, but rather the same erroneous concept used by other authors who also intervened through exercise, not articles from recognized specialists in hyperlipidemia, nor from world-renowned societies (National Cholesterol Education Panel (NCEP), Adult Treatment Panel III (ATP III), European Atherosclerosis Society (EAS)...). They should keep in mind that a person who is heavier than the recommended weight is not necessarily obese; a person who has blood glucose values higher than desirable is not necessarily a diabetic. A person with less bone mineral density than desirable (normal) does not necessarily have osteoporosis. A person with less strength or muscle mass than expected does not necessarily have sarcopenia. A person who has cholesterol "or" triglycerides above the desirable values does not have dyslipidemia. There are some diagnostic criteria, some diagnostic cut-off points: there is pre-sarcopenia, there is osteopenia, there is overweight, etc. .... and they do not indicate which are the normative points they use for cholesterol and triglycerides. However, we should point out that for cholesterol, less than 200 is desirable, between 200 and 240 is borderline, and above 240 is hypercholesterolemia (ATPIII; NCEP, EAS...)..The study participants have slightly borderline values (211.33 and 221.35 mg/dL), not hypercholesterolemia (average values). Except that to maintain these borderline values, they need to take cholesterol-lowering agents, which you do not indicate whether or not you take, nor what drug you take (if they take).

R: We considered dyslipidemia the older women that had values of total cholesterol and triglycerides above normative values and used literature that supported our results. The normative points that you refer we used for total cholesterol and for triglycerides we used less that 150mg/dL is desirable, between 150-199 borderline and above 200 is high. Kopin and Lowenstein (2017; doi:10.7326/aitc201712050) referred that dyslipidemia is a major risk for CVD and in US from 2009 to 2012 more than 100million adults aged 20 years or older have levels of 200mg/dl or higher (above normative values) and this can contribute for the risk for ischemic cerebrovascular accident. Thus, we changed older women with dyslipidemia to older women with levels of total cholesterol and triglycerides above normal range (Line 105-106; 730-731).

There is an important bias, not stated in the document but in its response to this reviewer, since the training group were women who did want (did they have the conditions?) to participate in the training sessions, while the control group were women who did not want to participate in the training sessions. What were the reasons for not "wanting/powering" to participate?

R: The older women of the control group do not engage the exercise program because they did not wanted to exercise and/or to had the commitment to the program, they only wanted to do the assessments to know their results in every test.

The sample is poorly characterized, the origin of these people is not indicated, the data of the Bioethics Committee is still not provided, nor is the date or at least the code assigned by this committee to this research. In studies with clinical conclusions, neither the group mean nor the comparison of group means is sufficient: how many in each group improved and how many did not? The cut-off points for cholesterol as a risk factor are usually established in mmol/L. How many changed this value above the value that modifies the calculable risk factor? I don't understand table 3, I want to understand that what they are trying to expose with the statistical treatment is the ANOVA, the three moments of evaluation, but independently for each of the two groups (which is not the best), but the post hoc result is not pointed out (who is different from whom? BE vs BD, BE vs AD, BD vs AD), and they do not indicate what is "ES", I want to understand that it is the effect size, which in the methodology they say it is Cohen's, but what they have done (I think) is an ANOVA, so maybe the effect size is not and should not be Cohen's but others like partial eta squared. 

R: For each variable (i.e. triglycerides, total cholesterol), we performed an ANOVA mixed model to compare within and between groups over the time (the three moments of evaluation). Thus, we have modified this sentence accordingly in the statistical analysis section. Moreover, we indicate what is "ES" = Effect size in the legend of table 3 and correct the symbol for significant differences, + for *, in Column BD of EG (LINE 391-393). Lastly, we used Cohen's d for analyze the effect size between two moments of evaluation, such as before exercise (BE) and after detraining (AD).

They should show the values of all the physical tests that were performed, not just the delta, and compare whether there was a difference between the groups in each physical test and BMI.

R: We show all the values of all the assessments in Table 3.

The lack of clarity in the definition of hyperlipidemia, throughout the text they make hyperlipidemia synonymous with cholesterol above the desirable values (below clearly pathological values), so that they argue with articles that have studied the effect of physical exercise on hyperlipidemias (C-HDL, C-LCD, C-VDL,...) with their results on the supposed effect of exercise on cholesterol and triglycerides. For example the important reference they have introduced (ref. 13, Albarrati et al.) does not conclude as they say that physical exercise reduces cholesterol, the study concludes something different: "Tis study found that low- to moderate-aerobic exercise intensities did not reduce the levels of LDL except in few studies that have been limited to specifc populations...."

R: Albarrati et al. 2018, referred in their results that aerobic exercise of both low and moderate intensity resulted in a significant reduction of total cholesterol and also in their conclusion that reduce LDL levels in dyslipidemia population.

The studies included in the meta-analyses of the effect of physical exercise on hyperlipidemia, in the vast majority have studied lipoproteins; and the effects of exercise are focused on these lipoproteins, which are clearly related to cardiovascular risk, much more than cholesterol (and less so borderline cholesterol), (e.g., the effects of exercise on the lipoproteins that are clearly related to cardiovascular risk, much more than cholesterol (and less so borderline cholesterol):, (e.g.:  DOIs: 10.5551/jat.42937 , 10.1089/acm.2017.0104,   10.1038/sj.ejcn.1600784, 10.1161/JAHA.115.002014).

R: We agree. But in our study we only have access to total cholesterol, blood glucose and triglycerides tests.

The description of the intervention is still insufficient; with the data provided, other researchers would not be able to replicate this study: can individual intensity be prescribed in collective choreographies? What level of resistance did the elastics have? what resistance is "adequate" for each person? was the same elastic for the leg press exercise as for the lateral raise, or for the squats...? Can the whole session described be done in 45 minutes?

R: In collective choreographies the bpm of the songs is a pratical method to control the intensity of the exercise through the session. All resistance exercises were used in different circuits during the nine months and the sessions. For the band exercises every older women chose the band (from 3 different tension levels) for every exercise according to the adapted borg scale (CR10 – RPE). All the sessions had a duration of 45 minutes. (Line 319-321)

In short, it is a very interesting subject and of great utility for people whose health is threatened, and whose conclusions can be applicable to these people, so the designs of these studies must be impeccable, and the conclusions we reach must be well-founded and based on clear diagnostic concepts. This is why we must be extremely responsible and thorough in our research and in the review of our peers' research.

R: We thank you very much your suggestions and review process that improved our article.